# Development of Machine Learning Models for Prediction of Smoking Cessation Outcome

**DOI:** 10.3390/ijerph18052584

**Published:** 2021-03-05

**Authors:** Cheng-Chien Lai, Wei-Hsin Huang, Betty Chia-Chen Chang, Lee-Ching Hwang

**Affiliations:** 1Department of Medical Education, Taipei Veterans General Hospital, Taipei City 11217, Taiwan; kelvin2005s@gmail.com; 2Department of Family Medicine, Mackay Memorial Hospital 25160, Taipei City 11217, Taiwan; whh.5881@mmh.org.tw (W.-H.H.); betty905@gmail.com (B.C.-C.C.); 3Department of Medicine, Mackay Medical College, New Taipei City 25245, Taiwan

**Keywords:** smoking cessation, predictive model, machine learning, artificial neural network, precision medicine

## Abstract

Predictors for success in smoking cessation have been studied, but a prediction model capable of providing a success rate for each patient attempting to quit smoking is still lacking. The aim of this study is to develop prediction models using machine learning algorithms to predict the outcome of smoking cessation. Data was acquired from patients underwent smoking cessation program at one medical center in Northern Taiwan. A total of 4875 enrollments fulfilled our inclusion criteria. Models with artificial neural network (ANN), support vector machine (SVM), random forest (RF), logistic regression (LoR), k-nearest neighbor (KNN), classification and regression tree (CART), and naïve Bayes (NB) were trained to predict the final smoking status of the patients in a six-month period. Sensitivity, specificity, accuracy, and area under receiver operating characteristic (ROC) curve (AUC or ROC value) were used to determine the performance of the models. We adopted the ANN model which reached a slightly better performance, with a sensitivity of 0.704, a specificity of 0.567, an accuracy of 0.640, and an ROC value of 0.660 (95% confidence interval (CI): 0.617–0.702) for prediction in smoking cessation outcome. A predictive model for smoking cessation was constructed. The model could aid in providing the predicted success rate for all smokers. It also had the potential to achieve personalized and precision medicine for treatment of smoking cessation.

## 1. Introduction

Cigarette smoking is an important global health issue [1] and is a well-known modifiable risk factor for many diseases including cancer, cardiovascular diseases, respiratory diseases, malfunction of the reproductive system, and many other organ systems [2]. It is estimated that smoking and exposure to secondhand smoke reduced life expectancy by 15 years [3]. Smoking cessation has been proven to be beneficial in many aspects of human health, including decreasing the risk of lung cancers, other cancers, heart attack, stroke, and chronic lung disease [4]. In addition, risk of all-cause mortality can be reduced by 13% within the first five years of smoking cessation [5]. More than one quarter of adult smokers are making attempts to quit smoking [6]. Assisting patients in quitting smoking is thus an important task for healthcare providers.

Evidence-based treatment and guideline for assistance in smoking cessation had been proposed and promoted [7]; however, only less than one third of the participants could achieve the goal of abstinence [8]. Many physicians found counseling for smoking cessation ineffective and time-consuming [9], and did not routinely do so in daily practice. To overcome this problem, several factors had been proposed to identify smokers who had a better chance of quitting, including the level of nicotine dependence, exhaled carbon monoxide (CO) concentration, cigarette amount per day, the age at smoking initiation, previous quit attempts, marital status, emotional distress, temperament and impulsivity scores, and the motivation to stop smoking [10,11,12,13]. However, individual use of these factors for prediction could lead to conflicting results that were not straightforward enough for the physicians and patients to interpret and apply. Providing a prediction model might be a favorable way to understand the chance of quitting smoking for each individual smoker. 

Health outcome prediction models had been developed using methods of machine learning over recent years [14,15]. Some examples included prediction of postoperative in-hospital mortality [16], complications in patients with diabetes mellitus [17], and occurrence of cardiovascular diseases in patients on dialysis [18]. For smoking cessation, a decision tree model developed with machine learning to predict smoking cessation treatment outcome was proposed by Coughlin et al in 2018 [19]. The study included 161 participants, with 90 in the training dataset and 71 in the validation dataset, yielding an average correct classification rate of about 64%. However, a prediction model for smoking cessation constructed with a larger dataset is still lacking.

In this study, we aimed to construct a prediction model for smoking cessation using machine learning algorithms with a larger dataset to give more informative and more reliable results. Input features included parameters that could be readily collected at the first visit of the patient, so the model could be applied without difficulties by the physicians and the patients. The output of the model would be a prediction of success rate for smoking cessation. With a predicted success rate, shared decision making could be made more easily with patients who wished to quit smoking.

## 2. Materials and Methods

### 2.1. Data Acquirement

Data of patients enrolled in the smoking cessation program between 2010 and 2018 in a medical center in northern Taiwan was reviewed. A full course of the program lasted for 8 weeks, and each patient could enroll in the program at most twice a year, based on the regulations of Taiwan Health Promotion Administration. Each treatment course consisted of several visits to the outpatient clinics of a physician or a health educator. Exhaled CO level was obtained at each visit. Physicians would prescribe medication for patients based on clinical judgments. We inquired about the abstinence status of the patient 3 months and 6 months after the first visit by telephone. Each enrollment was viewed as a set of independent data. This study was approved by the Ethics Committee of Mackay Memorial Hospital (Institutional review board number: 17MMHIS049).

Out of the 7424 enrollments reviewed in the study, 2549 were excluded due to lack of data on abstinence status, other missing data, incorrectly recorded value, or a recorded body weight out of the desirable range of 30 kg to 150 kg. The range was set to exclude extreme outliers that may decrease the discrimination ability of the specific feature in the model. The remaining 4875 enrollments were included for further analysis. The flowchart of data inclusion and preprocessing was shown in Figure 1.

### 2.2. Feature Selection and Data Preprocessing

Input features applied for the model included: sex, age, body weight, duration of smoking, daily amount of cigarette, motivation for smoke cessation, counseling done by health educator or physician, total score of the Fagerstrom Test for Nicotine Dependence (FTND) questionnaire [20], as well as the individual score of the 6 items in the questionnaire, conduction of CO test and the exhaled CO level, medication prescribed at first visit (no medication, nicotine replacement therapy (NRT), bupropion, or varenicline), and use of varenicline during the course. The 6 items in the FTND questionnaire included (1) Number of cigarettes smoked per day: 10 or less = 0; 11–20 = 1; 21–30 = 2; 31 or more = 3. (2) Time to first cigarette of the day: 60 min or more = 0; 31–60 min = 1; 6–30 min = 2; 0–5 min = 3. (3) Having difficulty not smoking in no-smoking areas: No = 0; Yes = 1. (4) Which cigarette would be the most difficult to give up? First in the morning = 1; Others = 0. (5) Smoke more frequently in the first hours after waking: No = 0; Yes = 1. (6) Still smoke when ill in bed: No = 0; Yes = 1.

We chose to include these predictors in our model based on that (1) It showed the patient’s baseline characteristics, (2) It had been proven to be an independent predictor in previous studies, and (3) It was the treatment the patient received. It is likely that including well-proven predictors provides more information to the outcome, thus improves the model performance.

The primary outcome of the model was a binary variable (1, 0) defined as the final abstinence status available from the patient within 6 months. If a patient had reported the abstinence status at the time at 6-month follow up, the result of 6-month follow up would be used. On the other hand, if a patient had not reported the abstinence status at 6-month follow up but had reported at 3 month follow-up, the result of 3-month follow up would be used. If a patient did not answer both times, the data would be excluded. Out of the 4875 enrollments, 3680 (75.5%) of the abstinence status was obtained from the 6-month period, and 1195 (24.5%) was obtained from the 3-month period.

The data was randomly divided into 4375 enrollments in the training dataset and 500 enrollments in the testing dataset. We limited the testing dataset to 500 samples in order to reserve more samples to be used for model training, and also ensure that the number of testing data was adequate for statistical analysis. The 4375:500 split was close to 90:10, and it was common to apply a split of either 70:30, 80:20, or 90:10 for construction of machine learning models [16,21,22]. For the artificial neural network (ANN), support vector machine (SVM), and k-nearest neighbor (KNN) models, the training data was rescaled to the range of 0 to 1 in every feature, and the testing data was rescaled according to the rescale index of the training data.

### 2.3. Machine Learning Model Development

In our study, we tried to develop a machine learning model to predict the probability of smoking cessation with features available at first visit. The candidate algorithms included: ANN, SVM, random forest (RF), logistic regression (LoR), KNN, classification and regression tree (CART), and naïve Bayes (NB).

The ANN model was made with Python 3.7 (Python Software Foundation, Wilmington, DE, USA) using Tensorflow 1.14.0 (Google Brain Team, Mountain View, CA, USA). The ANN was designed as a feedforward network with 4 fully-connected layers: the input layer consisted of 20 nodes, the hidden layer one consisted of 10 nodes, the hidden layer two consisted of 5 nodes, and the output layer consisted of 2 nodes. One-hot-encoding was used for the output layer, so that the probability of each categorical output could be calculated. For each connection between layers, the weight function was initialized randomly with normal distribution, and the bias function was initialized with zeros. The loss function of the model was defined as cross entropy. During the training process of the model, the training dataset would be given to the ANN, and the ANN would learn by optimizing the weight and bias between the connections, in order to minimize the loss function, at a learning rate optimized with adaptive moment estimation. To avoid overfitting of the training dataset, early stopping and a dropout rate of 0.4 were applied. 

For other methods, including SVM, RF, LoR, KNN, CART, and NB, the training process was achieved with Python 3.7 using Scikit-learn 0.21.2 [23]. For SVM, RF, KNN, and CART, a 20% validation dataset was derived randomly from the training dataset for each training section. Hyperparameters of the models were adjusted aggressively with experiments to achieve the best performance for the validation dataset. We prevented overfitting by limiting the complexities of the models. The final adopted SVM model was using linear classifier as kernel function; the RF model held 100 trees with maximum features and maximum depth set to 8; the KNN model was set to 81 neighbors. For LoR and NB, the whole training dataset was used due to the fixed results and the absence of adjustable hyperparameters.

### 2.4. Statistical Analysis

The testing dataset was used for further statistical analysis of the performance of the machine learning models. For each model, the result of the output was transferred to a number between 0 to 1 indicating the predicted success rate for smoking cessation. By design, the best performance of the models would be given at the cutoff point 0.5, which meant that any output value greater than or equal to 0.5 would be considered a positive prediction (or abstinence in the study), and any output value less than 0.5 would be considered a negative prediction (or non-abstinence in the study). By adjusting the cutoff point, different combinations of sensitivity and specificity could be achieved. Hence, an ROC curve could be drawn by moving the cutoff point from 0 to 1. The sensitivity, specificity, accuracy, and the ROC value were calculated to examine the performance of the machine learning models. The calculation of the CI for the ROC values and the comparison between different ROC values were performed with MedCalc 19.2 using the method proposed by DeLong et al [24].

### 2.5. Application of the Machine Learning Model

An application was made for physicians and patients to utilize the machine learning model. Through entering or changing the input values, the predicted success rate for smoking cessation would change accordingly.

## 3. Results

### 3.1. Characteristics of Enrolled Data

Of the 4875 enrollments including 3835 (78.7%) men and 1040 (21.3%) women, the average age was 46.7 years old and the average duration of smoking was 25.0 years. Over the eight-week period of the smoking cessation program, the average number of visits was 2.1 ± 1.1. The abstinence rate was 53.6% (*n* = 2675). The 4875 enrollments were randomly split into 4375 in the training dataset and 500 in the testing dataset. The abstinence rate was 53.7% in the training dataset and 53.4% in the testing dataset (*p* = 0.9092). Comparison of characteristics between the training and testing dataset were shown in Table 1. The statistics for homogeneity testing were shown in Appendix A.

### 3.2. Model Performance

After training the machine learning models with the training dataset, the testing dataset was used to test for the performance. The sensitivities, specificities, accuracies, and ROC values were shown in Table 2. To compare the ROC values of different models, the *p* value between every 2 models was calculated and presented in Table 3 (presented with *p* value comparing the 2 models labeled by the corresponding column and row).

By observing the ROC values of the models, there was a trend that the ANN, SVM, RF, and the LoR models had a better performance compared with the KNN, CART, and the NB models, though many of the comparisons were not statistically significant. Of the four better models, the ANN model had the best accuracy of 0.640 and an ROC value of 0.660 (95% CI: 0.617–0.702). The sensitivity and specificity of the ANN model was 0.704 and 0.567 respectively. As a result, the ANN model was adopted as the desired model for smoking cessation clinic setting in the study. The ROC curves of different models were shown in Figure 2, and the ROC curve of the ANN model with 95% CI boundary was shown in Figure 3.

## 4. Discussion

In our study, we constructed predictive models using seven different machine learning methods to predict the success rate of smoking cessation for current smokers with data available at the first visit. While the comparison of ROC between different models mostly revealed no statistical significance, the ANN model yield a better accuracy and ROC value, and was thus adopted as the desired model in the study.

Previous works on this topic focused mainly on identifying the independent predictors and describing the odds ratio for each predictor [11]. However, patients might host varying combinations of these predictors, which would lead to conflicting results and difficult interpretation for each individual in clinical practice. This situation could be avoided with the aid of prediction models. An attempt of constructing a classification model with machine learning on this topic was done by Coughlin et al. [19] in 2018. A decision tree model was made with an average correct classification rate of 64% of the full tree and 74% of the first split of tree in the validation cohort, and 81% of the full tree and 70% of the first split of tree in the training cohort, using delay discounting as the first split of the tree. The results were inspiring but should be interpreted carefully that overfitting might occur in a relatively small database, and the same protocol should be used in both the training and the validation cohort. In our study, a larger database (*n* = 4875) was used, and several different machine learning algorithms were applied. With the algorithms of machine learning, especially ANN, more flexible modeling could be achieved, with more complex pattern between the inputs and outputs identified [25]. 

The performance of our ANN model reached an AUC of 0.660, with a sensitivity of 0.704 and a specificity of 0.567. While an AUC > 0.7 was more preferable when analyzing examination tools [26], more work would be needed to achieve better AUC on predicting smoking cessation, including adding more features to the input. In our study, the 20 input features mainly consisted of the patient’s physical status, smoking status, and the intervention provided. These features were chosen in consideration of their easy accessibility, and most of these features were previously proven to be independent predictors in smoking cessation. However, these information may contribute only to a limited portion of the final abstinence status of the patient. Other factors might also be important, including the patient’s socioeconomic status, education level, family support, specific motivation for quitting, previous quit attempts, marital status, emotional distress, and the executive function and impulsivity measures. Addition of more related factors and recruitment of more data for training would be possible ways to further improve the performance of the model. However, adding too much input features also increases the barrier of applying the model in a clinical setting. The balance between model performance and convenience of use should be considered. Inter-correlation between selected features is a concern when more features are included. A correlation matrix of the input features in our study was provided in Appendix A. In this study, we considered 20 input features not being very much compared with previous studies [14,16,18], and the algorithms constructed in our study were not very complex. Since the models were relatively simple, the influence of inter-correlation from data should be conquered during model training, and would not cast a major problem to the performance.

An advantage of using the model is to understand the individual impact of change for a certain factor on smoking cessation. For example, losing 5 kg of weight would increase the probability of quitting by 10% in one patient, while for another patient, losing weight might not have the same effect, but changing the medication from NRT to varenicline would increase that patient’s probability of quitting by 15%. This example demonstrates that good prediction models provide keys to personalized and precision medicine. This would also encourage the physician to find better ways of quitting for the patients, thereby increasing the patient’s confidence, which is also an important component in successful smoking cessation. Further studies would be needed to confirm the effect of our prediction model in the clinical setting.

There were some limitations in the study. First, the outcome, the last available abstinence status within six months, was obtained through self-report by the patient. This could cause bias and misclassification. For an objective outcome measurement, exhaled CO [27,28] and urine cotinine level [29] could be used. In addition, 24.5% of the abstinence status was obtained from the three-month period, and 75.5% was from the six-month period. This time gap could cause bias for those who changed in abstinence status between the third and sixth month. To examine the effect of combining the three-month and six-month outcome to the prediction model, we rechecked the cessation rate in the original three-month and six-month report. The cessation rate was 55.6% at three month and 55.7% at six month. *p*-value of two-sample z test for cessation rate was 0.8887. Due to the similarity in cessation rate, the influence of adopting the combined outcome should be negligible.

Second, each enrollment was considered independent in our study. Repeated enrollments might cause bias in model construction and validation. To estimate the extent of influence of repeated enrollments in our study, we re-examined the 4875 data to look for data pairs with strong similarity. We tested for data pairs (calculated 11,880,375 pairs) that satisfied the following criteria: (1) age difference < ±3 years, (2) sex was the same, (3) body weight difference < ±2.5%, (4) ambition was the same, (5) duration of smoking < ±3 years, (6) cigarettes per day < ±10 sticks, (7) total FTND score < ±3 points, and (8) outcome was the same. Of the 11,880,375 pairs of data, 1781 pairs were found to be "similar" according to this criteria, which consisted of about 0.015%. With this level of similarity, the extent of bias caused by repeated enrollments might not be a major concern.

Third, other known and unknown predictors of smoking cessation might have been left out from our input features. However, if more features were to be included, the complexity of applying the model would become a barrier for clinical use. Furthermore, misreporting might occur during data collection of the input features, and we did not measure the compliance of the patients prescribed with medication. Lastly, the outcome of smoking cessation was related with race, culture, and health care policy. The model was based on medical practice in Taiwan and might not be able to be generalized to other countries worldwide.

## 5. Conclusions

A predictive model with ANN was constructed to predict the success rate of smoking cessation for current smokers using data available at the first visit. This model was easily applicable requiring only data collected at first patient visit. A predicted success rate could be provided for each patient, which could aid in shared decision making with the patient. Moreover, this method also had the potential to achieve personalized and precision medicine for treatment of smoking cessation.

## Figures and Tables

**Figure 1 ijerph-18-02584-f001:**
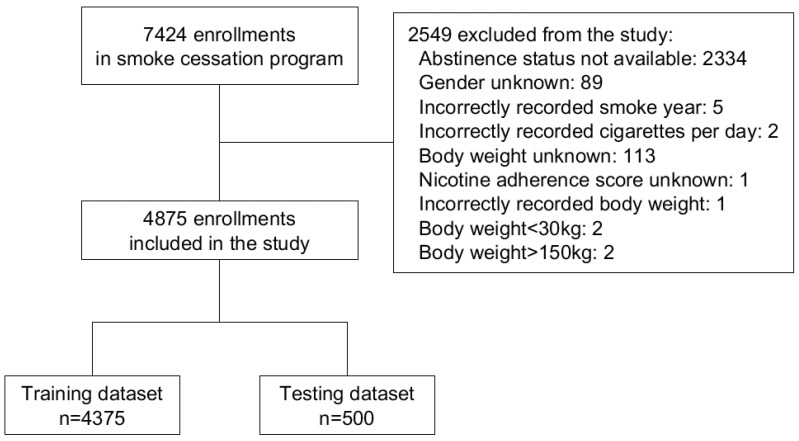
Flowchart of data inclusion and preprocessing.

**Figure 2 ijerph-18-02584-f002:**
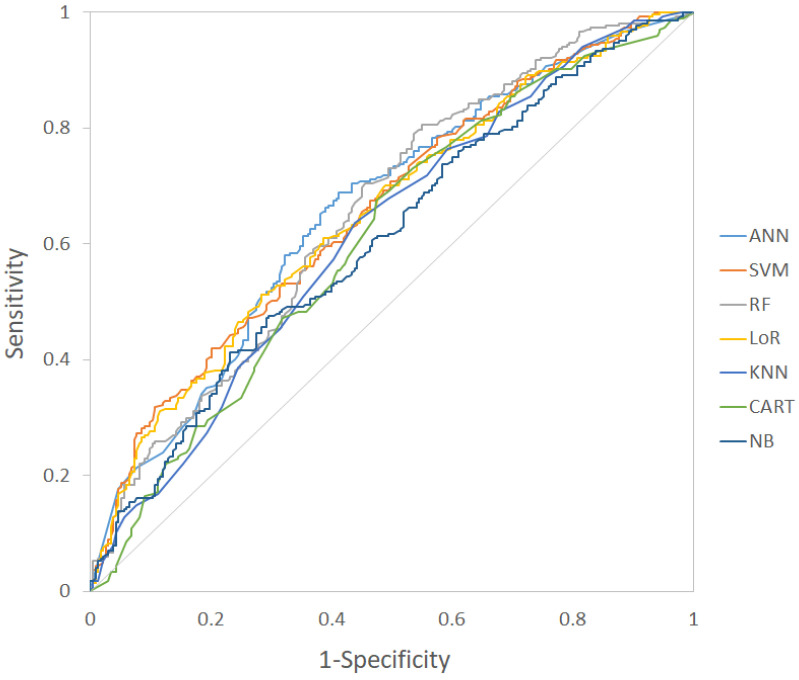
The ROC curves of different machine learning models for smoking cessation. ANN: Artificial neural network; SVM: Support vector machine; RF: Random forest; LoR: Logistic regression; KNN: K-nearest neighbors; CART: Classification and regression tree; NB: Naïve Bayes; ROC curve: Receiver operating characteristic curve.

**Figure 3 ijerph-18-02584-f003:**
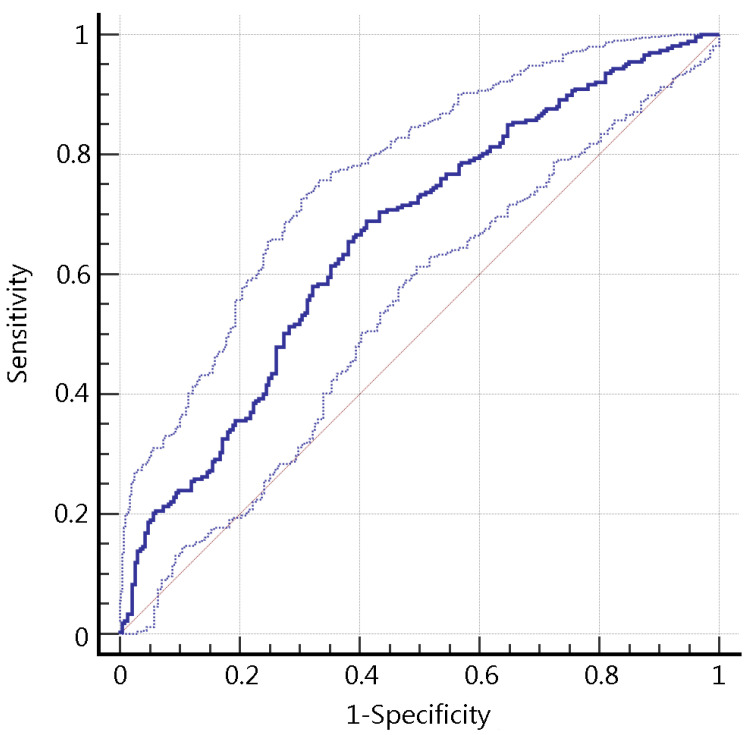
The ROC curve of the artificial neural network model, with 95% CI boundaries. ROC curve: Receiver operating characteristic curve; CI: Confidence interval.

**Table 1 ijerph-18-02584-t001:** Baseline characteristics of the training and testing dataset.

		Total (*n* = 4875)	Training Dataset (*n* = 4375)	Testing Dataset (*n* = 500)	*p*-Value
Gender (*n*, %)	Female	1040 (21.3)	938 (21.4)	102 (20.4)	0.5907
	Male	3835 (78.7)	3437 (78.6)	398 (79.6)	
Age (years)		46.7 ± 12.7	46.7 ± 12.7	46.9 ± 13.0	0.8325
Body weight (kg)		71.0 ± 14.9	71.1 ± 15.0	70.6 ± 14.3	0.5200
Duration of smoking (years)		25.0 ± 12.2	25.1 ± 12.2	24.7 ± 12.2	0.5294
Number of cigarettes smoked per day at baseline (stick)		20.1 ± 12.3	20.1 ± 12.4	20.2 ± 11.9	0.8987
Ambition (urge to quit)	Yes	2692 (55.2)	2430 (55.5)	262 (52.4)	0.1806
	No	2183 (44.8)	1945 (44.5)	238 (47.6)	
Physician clinics visit	Yes	4393 (90.1)	3937 (90.0)	456 (91.2)	0.3899
	No	482 (9.9)	438 (10.0)	44 (8.8)	
Educator clinics visit	Yes	1426 (29.3)	1289 (29.5)	137 (27.4)	0.3368
	No	3449 (70.7)	3086 (70.5)	363 (72.6)	
FTND score (point)		6.4 ± 2.3	6.4 ± 2.3	6.2 ± 2.4	0.1400
Exhaled CO level tested at baseline	Yes	3987 (81.8)	3576 (81.7)	411 (82.2)	0.7995
	No	888 (18.2)	799 (18.3)	89 (17.8)	
Exhaled CO levels (ppm)		15.9 ± 10.1	15.9 ± 10.1	16.0 ± 10.3	0.7581
Smoking cessation drugs prescribed at the 1st visit	Nil	441 (9.1)	399 (9.1)	42 (8.4)	0.1406
	NRT	943 (19.3)	846 (19.3)	97 (19.4)	
	Bupropion	9 (0.2)	6 (0.1)	3 (0.6)	
	Varenicline	3482 (71.4)	3124 (71.4)	358 (71.6)	
Use varenicline during treatment	No	1338 (27.4)	1204 (27.5)	134 (26.8)	0.7325
	Yes	3537 (72.6)	3171 (72.5)	366 (73.2)	
Point prevalence abstinence (*n*, %)	Success	2615(53.6)	2348 (53.7)	267 (53.4)	0.9092
	Fail	2260(46.4)	2027 (46.3)	233 (46.6)	

FTND: Fagerström Test of Nicotine Dependence; CO: Carbon monoxide; NRT: Nicotine Replacement Therapy.

**Table 2 ijerph-18-02584-t002:** Performance of different machine learning models for prediction of smoking cessation outcome.

	Sensitivity	Specificity	Accuracy	ROC Value (95% CI)
ANN	0.704	0.567	0.640	0.660 (0.617–0.702)
SVM	0.768	0.433	0.612	0.658 (0.614–0.699)
RF	0.757	0.485	0.626	0.654 (0.610–0.695)
LoR	0.742	0.459	0.608	0.653 (0.609–0.694)
KNN	0.764	0.408	0.598	0.618 (0.573–0.660)
CART	0.674	0.528	0.606	0.612 (0.568–0.655)
NB	0.614	0.524	0.568	0.608 (0.564–0.651)

ANN: Artificial neural network; SVM: Support vector machine; RF: Random forest; LoR: Logistic regression; KNN: K-nearest neighbors; CART: Classification and regression tree; NB: Naïve Bayes; ROC value: Receiver operating characteristic value; CI: Confidence interval.

**Table 3 ijerph-18-02584-t003:** Comparison of ROC values between different machine learning models for smoking cessation.

	ANN	SVM	RF	LoR	KNN	CART	NB
ANN	1.0000						
SVM	0.7997	1.0000					
RF	0.6882	0.8158	1.0000				
LoR	0.4873	0.2595	0.9518	1.0000			
KNN	0.0491	0.0601	0.1308	0.0945	1.0000		
CART	0.0505	0.058	0.0615	0.0944	0.8391	1.0000	
NB	0.0068	0.0009	0.0335	0.0031	0.6769	0.8865	1.0000

The results were shown in *p* value. ANN: Artificial neural network; SVM: Support vector machine; RF: Random forest; LoR: Logistic regression; KNN: K-nearest neighbors; CART: Classification and regression tree; NB: Naïve Bayes.

## Data Availability

Data were available from the Outpatient Smoking Cessation Treatment Database established by Taiwan Health Promotion Administration. Due to legal restrictions imposed by the government of Taiwan in relation to the “Personal Information Protection Act”, data cannot be made publicly available.

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
