# Peer review of "Development of Machine Learning Models for Prediction of Smoking Cessation Outcome"

_ijerph, 2021, doi:10.3390/ijerph18052584_

Round 1
Reviewer 1 Report
The manuscript is well written and interesting to read. The contents are understandable to the reader, even if not familiar with machine learning.
A prediction tool for the probability of success of smoking cessation interventions based on machine learning is interesting for clinical routine. The large data set is huge strength of this study. The authors compare different approaches and can identify a most promising model. However, there are some statistical and methodological issues, which should be improved. Detailed comments are given below.
L 21: „A reliable predictive model for smoking cessation was constructed “
The authors should phrase such a strong conclusion a little bit more carefully.
L35 -36:
Maybe the authors refer to unsuccessful attempts in smoking cessation. If yes, please point this out.
L115 – 118:
This sentence reads, as if all predictors included in the particular models were rescaled to a binary value [0,1]. For categorical variables, the authors seem to have used one-hot-encoding, but how did the authors handle continuous variable like the exhaled CO levels? Did the authors apply threshold values for a positive or negative prediction? Please explain this issue to the reader.
L 157:
Please name the specific method applied for the calculation of the CI.
L 145-158:
Later on, in the results section, the authors compare the ROC-values of the different models and provide p-values. Please describe which test the authors applied to obtain the p-values and provide the particular test-values. The information of p-values without the associated test value is relatively limited. In addition: Did the authors correct for multiple testing?
Figure 2 and 3:
Please use identically scales for x-axis (and respectively y-axis) in both figures.
L195-210:
The authors discuss the influence of the number of predictors on the AUC, the sensitivity and the specificity of the model. Further, they discuss more influence exerting factors, which should be included in future, analysis. It is right, that an increasing number of predictors reduce the acceptance of the tool in clinical routine. Additionally, an increasing number of predictors, especially if redundant, influence the accuracy of a model. Therefore, I recommend examining the inter-correlation of actual and future predictors. Maybe highly correlated predictors are redundant. Additionally a good tool to examine which predictors should be included in a model are directed acyclic graphs. Freeware like DAGITY (http://www.dagitty.net/) could be helpful.
In any case, the authors should discuss the inter-correlation aspect of the predictors for their models.
L224 – 228:
The authors are right, that their model could be biased by the mixing of 3-month and 6-month period results. The authors should give the information to reader, how many of the 4875 results are from 3-month reports and from 6-month reports. Maybe the authors could obtain enough cases for adequate enrollments for machine learning only using 6-month reports. Please include this information in the description of the cohort and discuss the proportion of both in the discussion section. One point should be, that the probability of restart smoking after 3- and 6 month of cessation.

Author Response
Response to Reviewer 1 Comments
Thank you for your letter and constructive comments concerning our manuscript entitled “Development of Machine Learning Models for Prediction of Smoking Cessation Outcome”. We have studied your comments carefully and made corresponding revision which we hope to meet your suggestions. The response to your questions or comments were shown in details in the following texts:
Point 1: L 21: „A reliable predictive model for smoking cessation was constructed “
The authors should phrase such a strong conclusion a little bit more carefully.
Response 1: Thank you for your suggestion. We have toned down our conclusion in the abstract and on Lines 21-22 (A predictive model for smoking cessation was constructed), and in the manuscript on Lines 175-177 (In our study, we constructed predictive models using 7 different machine learning methods to predict the success rate of smoking cessation for current smokers with data available at the first visit).
Point 2: L35-36: Maybe the authors refer to unsuccessful attempts in smoking cessation. If yes, please point this out.
Response 2: Thank you for the comment. We have clarified the statement on Lines 35-36 (More than one quarter of adult smokers are making attempts to quit smoking[6]). The unsuccessful attempts in smoking cessation is discussed in the next paragraph on Lines 38-40 (Evidence-based treatment and guideline for assistance in smoking cessation had been proposed and promoted[7], however, only less than one third of the participants could achieve the goal of abstinence[8]).
Point 3: L115-118: This sentence reads, as if all predictors included in the particular models were rescaled to a binary value [0,1]. For categorical variables, the authors seem to have used one-hot-encoding, but how did the authors handle continuous variable like the exhaled CO levels? Did the authors apply threshold values for a positive or negative prediction? Please explain this issue to the reader.
Response 3: Thank you for the comment. We have clarified the statement on Lines 115-118 (For the artificial neural network (ANN), support vector machine (SVM), and k-nearest neighbor (KNN) models, the training data was rescaled to the range of 0 to 1 in every feature, and the testing data was rescaled according to the rescale index of the training data). For categorical variables, we used one-hot-encoding and they would not be rescaled because they were already normalized to 0-1 by nature. Continuous variables were normalized to the range of 0 to 1 based on the data from the training data. (For testing data, the range would not be 0 to 1 because the largest and the smallest number in the 2 dataset would not be the same).
Point 4: L 157: Please name the specific method applied for the calculation of the CI.
Response 4: Thank you for the comment. For the statistical analysis of ROC values, the method provided by DeLong et al (DeLong ER, DeLong DM, Clarke-Pearson DL. Comparing the areas under two or more correlated receiver operating characteristic curves: a nonparametric approach. Biometrics. 1988;44(3):837-45.) was adopted, in which a covariance matrix estimator of theta was derived for ROCs and made the calculation of CI and nonparametric comparison of two ROCs possible. We used MedCalc 19.2 for the analysis with this specific method. We have revised the manuscript to clarify the statement on Lines 156-158 (The calculation of the CI for the ROC values and the comparison between different ROC values were performed with MedCalc 19.2 using the method proposed by DeLong et al[23]).
Point 5: L 145-158: Later on, in the results section, the authors compare the ROC-values of the different models and provide p-values. Please describe which test the authors applied to obtain the p-values and provide the particular test-values. The information of p-values without the associated test value is relatively limited. In addition: Did the authors correct for multiple testing?
Response 5: Thank you for the comment. We adopted the method proposed by DeLong et al for ROC comparison. In table 3, the p-value implicates the presence or absence of statistical difference in the performance between 2 different models. To correctly interpret the results, problem of multiple testing should be considered. Although most of the comparisons were not significant in table 3, hence we did not further discuss the problem of multiple testing in our manuscript, a trend could be seen that the models in the upper part of table 3 performed better than the models in the lower part.
Point 6: Figure 2 and 3: Please use identically scales for x-axis (and respectively y-axis) in both figures.
Response 6: Thank you for the suggestion, Figure 3 was revised and re-uploaded.
Point 7: L195-210: The authors discuss the influence of the number of predictors on the AUC, the sensitivity and the specificity of the model. Further, they discuss more influence exerting factors, which should be included in future, analysis. It is right, that an increasing number of predictors reduce the acceptance of the tool in clinical routine. Additionally, an increasing number of predictors, especially if redundant, influence the accuracy of a model. Therefore, I recommend examining the inter-correlation of actual and future predictors. Maybe highly correlated predictors are redundant. Additionally a good tool to examine which predictors should be included in a model are directed acyclic graphs. Freeware like DAGITY (http://www.dagitty.net/) could be helpful. In any case, the authors should discuss the inter-correlation aspect of the predictors for their models.
Response 7: Thank you for the delicate suggestion. Several aspects of input feature were considered in our models, including the patient’s physical status (gender, age, body weight), the patient’s smoking status (duration, number of cigarettes per day, FTND score parameters, CO level, urge to quit), and the intervention the patient received (visit to physician clinics, visit to educator clinics, medication use). Most of the adopted features were previously studied and proven to be independent predictors in smoking cessation. We have included the reasons for the features selected on Lines 200-202 (These features were chosen in consideration of their easy accessibility, and most of these features were previously proven to be independent predictors in smoking cessation); also, the discussion of the effect of inter-correlation was included on Lines 211-213 (Inter-correlation between selected features is a concern when more features were included. Recruitment of redundant input features would influence the training process and the final performance of the models, for which the degree of influence differs between different machine learning algorithms).
Point 8: L224 – 228: The authors are right, that their model could be biased by the mixing of 3-month and 6-month period results. The authors should give the information to reader, how many of the 4875 results are from 3-month reports and from 6-month reports. Maybe the authors could obtain enough cases for adequate enrollments for machine learning only using 6-month reports. Please include this information in the description of the cohort and discuss the proportion of both in the discussion section. One point should be, that the probability of restart smoking after 3- and 6 month of cessation.
Response 8: Thank you for the comment. In our study, we aimed to predict the final available outcome within 6 months as our target, of which 3,689 of the abstinence status was obtained from the 6-month period, and 1,195 (24.5%) was obtained from the 3-month period. We have revised and presented the information on Lines 107-109 (Out of the 4,875 enrollments, 3,680 (75.5%) of the abstinence status was obtained from the 6-month period, and 1,195 (24.5%) was obtained from the 3-month period). We also revised the discussion section on Lines 227-231 to include this information as our limitation (In addition, 24.5% of the abstinence status were obtained from the 3-month period, and 75.5% were from the 6-month period. The time gap might cause bias for those who changed in abstinence status between the third and sixth month. Furthermore, we could not obtain the abstinence status and patients might restart smoking after the 6-month period).

Reviewer 2 Report
This is an interesting study involving a large number of patients from a smoking cessation program. I have few comments:
Major comments: Investigators need to tone down the conclusion regarding the clinical utility of the ANN model. A model with sensitivity of 0.704, a specificity of 0.567, and accuracy of 0.640 would have a limited value for predicting smoking cessation.
Minor comments: I assume that DT in table 3 is CART.
Author Response
Response to Reviewer 2 Comments
Thank you for your letter and constructive comments concerning our manuscript entitled “Development of Machine Learning Models for Prediction of Smoking Cessation Outcome”. We have studied your comments carefully and made corresponding revision which we hope to meet your suggestions. The response to your questions or comments were shown in details in the following texts:
Point 1: Major comments: Investigators need to tone down the conclusion regarding the clinical utility of the ANN model. A model with sensitivity of 0.704, a specificity of 0.567, and accuracy of 0.640 would have a limited value for predicting smoking cessation.
Response 1: Thank you for the suggestion. We have toned down our conclusion in the abstract and on Lines 21-22 (A predictive model for smoking cessation was constructed), and in the manuscript on Lines 175-177 (In our study, we constructed predictive models using 7 different machine learning methods to predict the success rate of smoking cessation for current smokers with data available at the first visit).
Point 2: Minor comments: I assume that DT in table 3 is CART.
Response 2: Thank you for the comment. We have revised the content in table 3.
Reviewer 3 Report
In this work the authors try to build a prediction model for smoking cessation using different machine learning algorithms. The authors considered a larger dataset with respect to previous attempts, by considering patients enrolled in the smoking cessation program in Taiwan between 2010 and 2018.
Input parameters can be readily collected at the first medical visit having the aim to provide a personalized prediction for smoking cessation.
The main results concern the achievement of the highest ROC values for the ANN model applied to the testing dataset, although all the used algorithms provided similar results.
In my opinion the study could have important implications, even if, as also stated by authors,it needs some improvements in order to be used universally by including additional important input parameters.
Limitations concern the lack of the used code as supporting materials and several flaws displayed by the English language: just the first line of the abstract is not correct.
Finally, the obtained p-values for statistical analysis usually refer to non-significant differences: the authors should discuss this evidence.
For the exposed reasons, I believe the manuscript can be accepted for publication on Int. J. Environ. Res. Public Health after performing the suggested minor corrections.
Author Response
Response to Reviewer 3 Comments
Thank you for your letter and constructive comments concerning our manuscript entitled “Development of Machine Learning Models for Prediction of Smoking Cessation Outcome”. We have studied your comments carefully and made corresponding revision which we hope to meet your suggestions. The response to your questions or comments were shown in details in the following texts:
Point 1: In this work the authors try to build a prediction model for smoking cessation using different machine learning algorithms. The authors considered a larger dataset with respect to previous attempts, by considering patients enrolled in the smoking cessation program in Taiwan between 2010 and 2018. Input parameters can be readily collected at the first medical visit having the aim to provide a personalized prediction for smoking cessation. The main results concern the achievement of the highest ROC values for the ANN model applied to the testing dataset, although all the used algorithms provided similar results. In my opinion the study could have important implications, even if, as also stated by authors, it needs some improvements in order to be used universally by including additional important input parameters. Limitations concern the lack of the used code as supporting materials and several flaws displayed by the English language: just the first line of the abstract is not correct.
Response 1: Thank you for your comment. We have fully analysed the results of the constructed models and related code in the study. Several English language issue were revised in the manuscript by a profession who is a native English speaker.
Point 2: Finally, the obtained p-values for statistical analysis usually refer to non-significant differences: the authors should discuss this evidence.
Response 2: Thank you for the suggestion. We have revised and clarify the reason we adopted the ANN model although most of the comparison between models showed no statistical significance. The revision could be seen on section 3.2. Model performance (By observing the ROC values of the models, there was a trend that the ANN, SVM, RF, and the LoR models had a better performance compared with the KNN, CART, and the NB models, though many of the comparisons were not statistically significant. Of the four better models, the ANN model had the best accuracy of 0.640 and an ROC value of 0.660 (95% CI: 0.617-0.702). The sensitivity and specificity of the ANN model was 0.704 and 0.567 respectively. As a result, the ANN model was adopted as the desired model for smoking cessation clinic setting in the study), and on Lines 175-179 (In our study, we constructed predictive models using 7 different machine learning methods to predict the success rate of smoking cessation for current smokers with data available at the first visit. While the comparison of ROC between different models mostly revealed no statistical significance, the ANN model yield a better accuracy and ROC value, and was thus adopted as the desired model of the study).

Reviewer 4 Report
The paper aims to predict the outcome of smoking cessation with machine learning models. I have a few suggestions that could help make the paper more clear for the readers, in my opinion:
- in "2.2 Feature selection and data preprocessing" you listed the feature without actually discussing the feature selection process.
In my opinion, would be interesting if you explain how the features were selected. Did you investigate the effect of removing some features on the prediction accuracy? Are the features approximately independent? This should be clarified, maybe a correlation matrix would be useful, to help the readers to understand and use the output of the model. - lines 135-137: That's how the training is supposed to work. Unless you want to quantify the "certain low level", I suggest removing this sentence because doesn't add information.
- In 2.3, the ANN model has been extensively described. A few more details on the other models would be useful.
- 3680 results were collected from the 6-months period and 1195 from the 3 months period. Would be interesting if you show how the classification performance and the smoking cessation outcome are distributed in the 2 datasets.
- As you said, repeated enrollments might cause bias in model construction. Were you able to know which data (or at least how many of them) were referred to repeated enrollments?
Author Response
Response to Reviewer 4 Comments
Thank you for your letter and constructive comments concerning our manuscript entitled “Development of Machine Learning Models for Prediction of Smoking Cessation Outcome”. We have studied your comments carefully and made corresponding revision which we hope to meet your suggestions. The response to your questions or comments were shown in details in the following texts:
Point 1: In "2.2 Feature selection and data preprocessing" you listed the feature without actually discussing the feature selection process. In my opinion, would be interesting if you explain how the features were selected. Did you investigate the effect of removing some features on the prediction accuracy? Are the features approximately independent? This should be clarified, maybe a correlation matrix would be useful, to help the readers to understand and use the output of the model.
Response 1: Thank you for the delicate suggestion. Several aspects of input feature were considered in our models, including the patient’s physical status (gender, age, body weight), the patient’s smoking status (duration, number of cigarettes per day, FTND score parameters, CO level, urge to quit), and the intervention the patient received (visit to physician clinics, visit to educator clinics, medication use). Most of the adopted features were previously studied and proven to be independent predictors in smoking cessation. We have included the reasons for the features selected on Lines 200-202 (These features were chosen in consideration of their easy accessibility, and most of these features were previously proven to be independent predictors in smoking cessation); also, the discussion of the effect of inter-correlation was included on Lines 211-213 (Inter-correlation between selected features is a concern when more features were included. Recruitment of redundant input features would influence the training process and the final performance of the models, for which the degree of influence differs between different machine learning algorithms).
Point 2: lines 135-137: That's how the training is supposed to work. Unless you want to quantify the "certain low level", I suggest removing this sentence because doesn't add information.
Response 2: Thank you for the suggestion. We have removed the sentence on Lines 135-137.
Point 3: In 2.3, the ANN model has been extensively described. A few more details on the other models would be useful.
Response 3: Thank you for the suggestion. The final adopted SVM model was using linear classifier as kernel function; the RF model held 100 trees with maximum features and maximum depth set to 8; the KNN model was set to 81 neighbors; for the CART model, the maximum features and maximum depth were also set to 8. In our study, the ANN model yield the better accuracy and ROC value, so we adopted it for the prediction in smoking cessation clinic setting. Due to the above reason, we described ANN more extensively in the methods section.
Point 4: 3680 results were collected from the 6-months period and 1195 from the 3 months period. Would be interesting if you show how the classification performance and the smoking cessation outcome are distributed in the 2 datasets.
Response 4: Thank you for the suggestion. It would be valuable to understand the difference of the model’s prediction performance and outcome between the 2 groups of data. However, data was de-identified during analyzation and could not be traced back. We aimed to predict the final available outcome within 6 months as our target.
Point 5: As you said, repeated enrollments might cause bias in model construction. Were you able to know which data (or at least how many of them) were referred to repeated enrollments?
Response 5: Thank you again for the valuable suggestion. We agreed that in prediction models, it is important to know and control repeated enrollments since this may cause bias. However, the data was de-identified during analyzation and could not be traced back. In the study, we viewed each enrollment independent and collected all the data again during every visit. Hence, even though repeated enrollment would cause bias, the extent of such bias would not be a major concern. We recognized this as our limitation.

Round 2
Reviewer 1 Report
Thank you for your detailed answers to mysugegstions. I recommend that the manuscript be accepted for publication in its present form.
Author Response
Response to Reviewer 1 Comments
Thank you for your letter and constructive comments concerning our manuscript entitled “Development of Machine Learning Models for Prediction of Smoking Cessation Outcome”. We have studied your comments carefully and made corresponding revision which we hope to meet your suggestions. The response to your questions or comments were shown in details in the following texts:
Point 1: Thank you for your detailed answers to my suggestions. I recommend that the manuscript be accepted for publication in its present form.
Response 1: We sincerely thank you for your constructive suggestions. We hope the manuscript could be improved and clarified for our readers.

Reviewer 4 Report
The paper is basically the same as the v1, with little minor adjustments.
Here are a few comments:
- Lines 211-213: "Inter-correlation between selected features is a concern when more features were included. Recruitment of redundant input features would influence the training process and the final performance of the models, for which the degree of influence differs between different machine learning algorithms" -> That's true. For this reason, a correlation matrix would be important in subsection 2.2
- "The final adopted SVM model was using linear classifier as kernel function; the RF model held 100 trees with maximum features and maximum depth set to 8; the KNN model was set to 81 neighbors". Consider adding this information in the model description, because can be useful for the readers.
- Again, would be interesting if you show how the classification performance and the smoking cessation outcome are distributed in the 3-months and 6-months datasets. Even if you didn't keep track of the data during the process, you should at least show how the cessation outcome is distributed in the original datasets.
- "In the study, we viewed each enrollment independent and collected all the data again during every visit. Hence, even though repeated enrollment would cause bias, the extent of such bias would not be a major concern. We recognized this as our limitation."
In this case, "Hence" seems not to be the appropriate word. How considering related events as unrelated should reduce the extent of the bias? Anyway, recognizing it as a limitation doesn't improve the reliability of the analysis. It's not necessary to investigate the accuracy of the model on the repeated enrollments, but at least should be checked if the dataset can be considered homogeneous. Otherwise, it would be better to work with fewer data.
Author Response
Response to Reviewer 4 Comments
Thank you for your letter and constructive comments concerning our manuscript entitled “Development of Machine Learning Models for Prediction of Smoking Cessation Outcome”. We have studied your comments carefully and made corresponding revision which we hope to meet your suggestions. The response to your questions or comments were shown in details in the following texts:
Point 1: Lines 211-213: "Inter-correlation between selected features is a concern when more features were included. Recruitment of redundant input features would influence the training process and the final performance of the models, for which the degree of influence differs between different machine learning algorithms" -> That's true. For this reason, a correlation matrix would be important in subsection 2.2
Response 1: Thank you for the suggestion. We provide the correlation matrix of the predictors as below (note that some predictors were divided into more than one variables for model input due to the catalogues nature of data; such as, for prescribed medication, there are 4 categories - no medication, nicotine replacement therapy, bupropion, and varenicline. It was transferred to 2 input features in our model, coding [0,0], [0,1], [1,0], [1,1]. For this reason, only 17 variables are presented in the table below).
Table. Pearson's correlation coefficient matrix of predictors |
|||||||||||||||||||||||||||
  |
Age |
Sex |
Weight |
Ambition |
CO level |
Therapy |
Cig per d |
Duration |
FTND 1 |
FTND 2 |
FTND 3 |
FTND 4 |
FTND 5 |
FTND 6 |
FTND score |
Doc OPD |
Edu OPD |
|
|||||||||
Age |
1.00000 |
0.06268* |
-0.10977* |
-0.00119 |
-0.11335* |
-0.01639 |
0.02727 |
0.83704* |
0.00727 |
-0.01718 |
-0.13481* |
-0.05886* |
-0.00033 |
0.05997* |
-0.02907* |
-0.05132* |
0.12280* |
|
|||||||||
Sex |
0.06268* |
1.00000 |
0.43658* |
0.00130 |
0.03895* |
0.03541* |
0.05312* |
0.16036* |
0.08190* |
-0.04099* |
-0.04654* |
-0.00576 |
-0.08166* |
-0.00282 |
-0.01779 |
-0.02319 |
0.03876* |
|
|||||||||
Weight |
-0.10977* |
0.43658* |
1.00000 |
0.00988 |
0.02796 |
0.04078* |
0.06782* |
-0.04123* |
0.10228* |
-0.00855 |
0.01103 |
-0.00713 |
-0.02225 |
0.01193 |
0.03437* |
0.00842 |
0.01988 |
|
|||||||||
Ambition |
-0.00119 |
0.00130 |
0.00988 |
1.00000 |
0.04031* |
0.17458* |
0.07031* |
0.00372 |
0.04363* |
0.06031* |
0.05614* |
0.06467* |
0.05399* |
0.06088* |
0.08834* |
0.24484* |
-0.17632* |
|
|||||||||
CO level |
-0.11335* |
0.03895* |
0.02796 |
0.04031* |
1.00000 |
0.03341* |
0.37728* |
-0.06504* |
0.28149* |
0.12030* |
0.08797* |
0.05514* |
0.19111* |
0.04846* |
0.24952* |
0.00471 |
0.06284* |
|
|||||||||
Therapy |
-0.01639 |
0.03541* |
0.04078* |
0.17458* |
0.03341* |
1.00000 |
0.06402* |
-0.00755 |
0.04846* |
0.03491* |
0.04356* |
0.03756* |
0.02631 |
0.06490* |
0.06727* |
0.48466* |
-0.19362* |
|
|||||||||
Cig per d |
0.02727 |
0.05312* |
0.06782* |
0.07031* |
0.37728* |
0.06402* |
1.00000 |
0.09285* |
0.60489* |
0.11548* |
0.14855* |
0.15917* |
0.24261* |
0.10052 |
0.44337* |
0.12384* |
-0.04805* |
|
|||||||||
Duration |
0.83704* |
0.16036* |
-0.04123* |
0.00372 |
-0.06504* |
-0.00755 |
0.09285* |
1.00000 |
0.08656* |
0.01430 |
-0.04858* |
0.00799 |
0.05679* |
0.06794* |
0.06461* |
-0.04772* |
0.11810* |
|
|||||||||
FTND 1 |
0.00727 |
0.08190* |
0.10228* |
0.04363* |
0.28149* |
0.04846* |
0.60489 |
0.08656 |
1.00000 |
0.13619* |
0.19526* |
0.22510* |
0.28855* |
0.12494* |
0.65227* |
0.06194* |
-0.08936* |
|
|||||||||
FTND 2 |
-0.01718 |
-0.04099* |
-0.00855 |
0.06031* |
0.12030* |
0.03491* |
0.11548* |
0.01430 |
0.13619* |
1.00000 |
0.22084* |
0.24702* |
0.40345* |
0.32974* |
0.57624* |
0.09246* |
-0.05805* |
|
|||||||||
FTND 3 |
-0.13481* |
-0.04654* |
0.01103 |
0.05614* |
0.08797* |
0.04356* |
0.14855* |
-0.04858* |
0.19526* |
0.22084* |
1.00000 |
0.34102* |
0.26020* |
0.26215* |
0.56110* |
0.11409* |
-0.10862* |
|
|||||||||
FTND 4 |
-0.05886* |
-0.00576 |
-0.00713 |
0.06467* |
0.05514* |
0.03756* |
0.15917* |
0.00799 |
0.22510* |
0.24702* |
0.34102* |
1.00000 |
0.23171* |
0.23121* |
0.55785* |
0.08809* |
-0.11871* |
|
|||||||||
FTND 5 |
-0.00033 |
-0.08166* |
-0.02225 |
0.05399* |
0.19111* |
0.02631 |
0.24261* |
0.05679* |
0.28855* |
0.40345* |
0.26020* |
0.23171* |
1.00000 |
0.25183* |
0.72978* |
0.05924* |
-0.05011* |
|
|||||||||
FTND 6 |
0.05997* |
-0.00282 |
0.01193 |
0.06088* |
0.04846* |
0.06490* |
0.10052* |
0.06794* |
0.12494* |
0.32974* |
0.26215* |
0.23121* |
0.25183* |
1.00000 |
0.52738* |
0.18663* |
-0.11567* |
|
|||||||||
FTND score |
-0.02907* |
-0.01779 |
0.03437* |
0.08834* |
0.24952* |
0.06727* |
0.44337* |
0.06461* |
0.65227* |
0.57624* |
0.56110* |
0.55785* |
0.72978* |
0.52738* |
1.00000 |
0.14807* |
-0.13889* |
|
|||||||||
Doc OPD |
-0.05132* |
-0.02319 |
0.00842 |
0.24484* |
0.00471 |
0.48466* |
0.12384* |
-0.04772* |
0.06194* |
0.09246* |
0.11409* |
0.08809* |
0.05924* |
0.18663* |
0.14807* |
1.00000 |
-0.51515* |
|
|||||||||
Edu OPD |
0.12280* |
0.03876* |
0.01988 |
-0.17632* |
0.06284* |
-0.19362* |
-0.04805* |
0.11810* |
-0.08936* |
-0.05805* |
-0.10862* |
-0.11871* |
-0.05011* |
-0.11567* |
-0.13889 |
-0.51515* |
1.00000 |
|
|||||||||
*, p<0.05.
As the table shows, many of the predictors have inter-correlations. Although these inter-correlation presents, we choose to include these predictors in our model based on that (1) It shows the patient’s baseline characteristics, (2) It has been proven to be an independent predictor in previous studies, and (3) It is the treatment the patient received. For baseline characteristics, we included the age, sex, and weight of the patient. Ambition, CO level, cigarettes per day, duration, and FTND scores were well proven independent predictors for success in smoking cessation. Visit to physician, visit to health educator, and the medication prescribed were 3 features that describe the treatments the patient received. We considered that, in training of machine learning models, whether the newly-added feature provide further information to the outcome would be an issue more important than examining the inter-correlation between features. It is likely that including well-proven predictors provides more information to the outcome, thus improves the model performance, although inter-correlation presents.
It may seem that 20 features are quite a lot for traditional model construction, but this is not the case in machine learning models. From our reference [14], models were built with 19, 30, and 46 features; from reference [16], an artificial neural network model was built with 87 features; model in reference [18] also host more than 20 input features. When using neural network for image recognition, although many complex programming are under development and the details are different with the algorithm used in our study, each pixel itself form an input X, and it is common to have more than 1000 input variables in the model. Back to our study, we consider 20 input feature not being very much, and the algorithm constructed in our study are not very complex. Since the models are relatively simple, the influence of inter-correlation from data should be conquered during model training, and should not cast a major problem to the performance.
Point 2: "The final adopted SVM model was using linear classifier as kernel function; the RF model held 100 trees with maximum features and maximum depth set to 8; the KNN model was set to 81 neighbors". Consider adding this information in the model description, because can be useful for the readers.
Response 2: Thank you for the suggestion. We have added the description on Lines 143-145.
Point 3: Again, would be interesting if you show how the classification performance and the smoking cessation outcome are distributed in the 3-months and 6-months datasets. Even if you didn't keep track of the data during the process, you should at least show how the cessation outcome is distributed in the original datasets.
Response 3: Thank you again for the suggestion. We have re-examined the original 3-month and 6-month report. The cessation rate from 3-month was 55.57%, and cessation rate from 6-month was 55.73%. P-value of 2-sample z test for cessation rate was 0.88866. We also tested our ANN model for the 2 datasets. The classification accuracy of the ANN model for the original 3-month and 6-month dataset was 65.00% and 65.65% respectively, with p-value of 0.54186. We would like to emphasise that the target of prediction in our study was defined to be the final available outcome within 6 months.
Point 4: "In the study, we viewed each enrollment independent and collected all the data again during every visit. Hence, even though repeated enrollment would cause bias, the extent of such bias would not be a major concern. We recognized this as our limitation."
In this case, "Hence" seems not to be the appropriate word. How considering related events as unrelated should reduce the extent of the bias? Anyway, recognizing it as a limitation doesn't improve the reliability of the analysis. It's not necessary to investigate the accuracy of the model on the repeated enrollments, but at least should be checked if the dataset can be considered homogeneous. Otherwise, it would be better to work with fewer data.
Response 4: Thank you again for the detailed comment. As stated in the response in round 1, we could not know which enrollment data are from the same person by currently available information. To estimate the extent of the influence of repeated enrollments in our study, we re-examine the 4,875 data to look for data pairs with strong similarity. That is, we test for data pairs (calculated 4,875 x 4,874 / 2 = 11,880,375 pairs) that satisfy the following criteria: (1) age difference < ±3 years (2) sex is the same (3) body weight difference < ±2.5% (4) ambition is the same (5) duration of smoking < ±3 years (6) cigarettes per day < ±10 sticks (7) total FTND score < ±3 points (8) outcome is the same. Of the 11,880,375 pairs of data, 1,781 pairs are found to be ‘similar’ according to this criteria, which consist of about 0.015%. With this level of similarity, we consider it to be safe to say that the extent of bias caused by repeated enrollments in our study should not be a major concern. In addition, we have compared the categorical and continuous variables in the training and testing dataset to test for homogeneity, and all results showed homogenous. The statistics for homogeneity testing are shown in the following table.
Table. Homogeneity testing between the training and test groups |
||
Categorical variables |
χ2 value |
p |
Sex |
0.2892 |
0.5907 |
Ambition |
1.7924 |
0.1806 |
Physician clinics visit |
0.7391 |
0.3899 |
Educator clinics visit |
0.9227 |
0.3368 |
Therapy |
0.1168 |
0.7325 |
Continuous variables |
F value |
p |
Age (years) |
1.04 |
0.5076 |
Body weight (kg) |
1.10 |
0.1567 |
Exhaled CO levels (ppm) |
1.04 |
0.5102 |
Number of cigarettes smoked per day at baseline (stick) |
1.09 |
0.2046 |
Duration of smoking (years) |
1.01 |
0.8982 |
FTND 1 |
1.01 |
0.9465 |
FTND 2 |
1.02 |
0.7583 |
FTND 3 |
1.00 |
0.9586 |
FTND 4 |
1.02 |
0.6997 |
FTND 5 |
1.09 |
0.1888 |
FTND 6 |
1.02 |
0.7223 |
FTND score (point) |
1.08 |
0.2665 |
Thank you very much again for your detailed review and valuable comments. We hope the responses above could meet your suggestion, and further clarify the study for our readers.

Round 3
Reviewer 4 Report
I appreciate the information provided in the Author Response. However, I think that these data should be made available to the readers, at least in the supplement (maybe better if in the paper), otherwise the review process is pointless.
Author Response
Response to Reviewer 4 Comments
Thank you for your letter and constructive comments concerning our manuscript entitled “Development of Machine Learning Models for Prediction of Smoking Cessation Outcome”. We have studied your comments carefully and made corresponding revision which we hope to meet your suggestions. The response to your questions or comments were shown in details in the following texts:
Point 1: I appreciate the information provided in the Author Response. However, I think that these data should be made available to the readers, at least in the supplement (maybe better if in the paper), otherwise the review process is pointless.
Response 1: Thank you for your detailed review and valuable comments.
First, we have further clarified the process and reason of input feature selection on Lines 102-106 in the Methods section (We chose to include these predictors in our model based on that (1) It showed the patient’s baseline characteristics, (2) It had been proven to be an independent predictor in previous studies, and (3) It was the treatment the patient received. It is likely that including well-proven predictors provides more information to the outcome, thus improves the model performance). We have also discussed the role of inter-correlation and provided the correlation matrix on Lines 217-223 in the Discussion section (Inter-correlation between selected features is a concern when more features are included. A correlation matrix of the input features in our study was provided in Supplementary Table 2. In this study, we considered 20 input features not being very much compared with previous studies[14, 16, 18], and the algorithms constructed in our study were not very complex. Since the models were relatively simple, the influence of inter-correlation from data should be conquered during model training, and would not cast a major problem to the performance).
Second, the comparison of cessation rate between the original 3- and 6-month data was included on Lines 240-244 in the Discussion section (To examine the effect of combining the 3-month and 6-month outcome to the prediction model, we rechecked the cessation rate in the original 3-month and 6-month report. The cessation rate was 55.6% at 3 month and 55.7% at 6 month. P-value of 2-sample z test for cessation rate was 0.8887. Due to the similarity in cessation rate, the influence of adopting the combined outcome should be negligible).
Lastly, the estimation of the impact of repeated enrollments was presented on Lines 246-254 in the Discussion section (To estimate the extent of influence of repeated enrollments in our study, we re-examined the 4,875 data to look for data pairs with strong similarity. We tested for data pairs (calculated 11,880,375 pairs) that satisfied the following criteria: (1) age difference < ±3 years (2) sex was the same (3) body weight difference < ±2.5% (4) ambition was the same (5) duration of smoking < ±3 years (6) cigarettes per day < ±10 sticks (7) total FTND score < ±3 points (8) outcome was the same. Of the 11,880,375 pairs of data, 1,781 pairs were found to be ‘similar’ according to this criteria, which consisted of about 0.015%. With this level of similarity, the extent of bias caused by repeated enrollments might not be a major concern).
